# Thyroid Hormone Withdrawal versus Recombinant Human TSH as Preparation for I-131 Therapy in Patients with Metastatic Thyroid Cancer: A Systematic Review and Meta-Analysis

**DOI:** 10.3390/cancers15092510

**Published:** 2023-04-27

**Authors:** Luca Giovanella, Maria Luisa Garo, Alfredo Campenní, Petra Petranović Ovčariček, Rainer Görges

**Affiliations:** 1Clinic for Nuclear Medicine and Molecular Imaging, Imaging Institute of Southern Switzerland, 6500 Bellinzona, Switzerland; 2Clinic for Nuclear Medicine, University Hospital of Zurich, 8091 Zurich, Switzerland; 3Fondazione Policlinico Universitario Campus Bio-Medico, Via Alvaro del Portillo, 200-00128 Roma, Italy; marilu.garo@gmail.com; 4Research Unit of Cardiac Surgery, Department of Cardiovascular Surgery, Università Campus Bio-Medico di Roma, Via Alvaro del Portillo, 21-00128 Roma, Italy; 5Nuclear Medicine Unit, Department of Biomedical and Dental Sciences and Morpho-Functional Imaging, University of Messina, 98122 Messina, Italy; acampenni@unime.it; 6Department of Oncology and Nuclear Medicine, University Hospital Center “Sestre Milosrdnice”, 10000 Zagreb, Croatia; p.petranovic@gmail.com; 7Clinic for Nuclear Medicine, University Hospital Essen, 45147 Essen, Germany; rainer.goerges@uni-due.de

**Keywords:** differentiated thyroid carcinoma, recombinant human-thyroid stimulating hormone, thyroid hormone withdrawal, ^131^I therapy

## Abstract

**Simple Summary:**

Differentiated thyroid carcinoma (DTC) is characterized by an excellent prognosis, with the exception of cases that develop in metastatic forms. Although I-131 has been shown to be an effective therapy in patients with metastatic DTC, whether its efficacy after recombinant human TSH (rhTSH) is comparable to endogenous TSH stimulation by thyroid hormone deprivation (THW) is still debated. Our present data indicated a lack of significant effect of rhTSH or THW pretreatment over the other on the effectiveness of I-131 therapy of metastatic DTC. This implies that concerns about the use of one or the other pretreatment should be deferred to clinical evaluations made considering patient characteristics and reduction in side effects.

**Abstract:**

Background. Differentiated thyroid carcinoma (DTC) is characterized by an excellent prognosis with a 10-year survival rate > 90%. However, when DTC develops in a metastatic form, it has been shown to significantly impact patient survival and quality of life. Although I-131 has been shown to be an effective therapy in patients with metastatic DTC, whether its efficacy after recombinant human TSH (rhTSH) is comparable to endogenous TSH stimulation by thyroid hormone deprivation (THW) is still debated. Our present study was prompted to compare clinical results obtained in metastatic DTC by I-131 administered after rhTSH and THW stimulation protocols, respectively. Methods. A systematic search on PubMed, Web of Science, and Scopus was performed from January to February 2023. Pooled risk ratios with 95% CI were determined for evaluating the initial response after to I-131 therapy after preparation with rhTSH or THW and the disease progression. To track the accumulation of evidence and reduce type I errors because of small data, a cumulative meta-analysis was performed. A sensitivity analysis was also performed to examine the impact of individual studies on overall prevalence results. Results. Ten studies were included with a total of 1929 patients pre-treated with rhTSH (n = 953) and THW (n = 976), respectively. The cumulative data of our systematic review and meta-analysis showed an increase in the risk ratio over the years without any change in favour of a pre-treatment or the other on the effectiveness of I-131 therapy of metastatic DTC. Conclusions. Our data suggest that pretreatment with rhTSH or THW has no significant impact on the effectiveness of I-131 therapy for metastatic DTC. This implies that concerns about the use of one or the other pretreatment should be deferred to clinical evaluations made considering patient characteristics and reduction in side effects.

## 1. Introduction

The worldwide incidence of thyroid cancer, of which differentiated thyroid carcinoma (DTC) is the most common and accounts for 90% of all thyroid carcinomas, has steadily increased in recent decades [1]. In general, DTC is characterized by slow disease progression in many cases and an excellent prognosis, with a 10-year survival rate > 90% [2]. However, when DTC is present or develops in a metastatic form, it has been shown to significantly impact patient survival and quality of life. Overall, about 30% of DTC patients have metastatic cancer, mostly at regional lymph nodes (five-year survival rate > 90%), while about 5% of DTC patients develop metastasis in distant organs such as the lungs and bone that are related to a significantly worse prognosis (i.e., five-year survival: 54.9%) [3,4]. Accordingly, likely due to the combination of higher incidence and increased mortality in metastatic DTC patients, an increased overall mortality rate has recently been reported in DTC patients [5]. Luckily, most patients with advanced, metastasized, DTC still have Iodine-131 (I-131)-avid metastatic lesions, and 40% of them achieve remission after I-131 treatment [6,7]. Moreover, a delay of six months in administration of I-131 therapy is associated with decreased five-year survival in patients with distant metastases [8]. Accordingly, in addition to TSH suppression [9], repeated courses of I-131 treatment are the standard of care to manage metastasized DTC as long as the disease remains iodine-avid [10,11].

Current guidelines recommend one to two weeks of low-iodine diet and adequate stimulation of the thyroid stimulating hormone (TSH) for an optimal preparation of patients to I-131 therapy [12]. In particular, increased TSH levels are required to enhance the expression and function of sodium-iodide symporter (NIS) and, in turn, to increase the absorbed radiation dose to target lesions [13]. Currently, TSH stimulation can be obtained by thyroid hormone withdrawal (THW) or recombinant human TSH (rhTSH) administration. Both methods proved to be equally effective for ablating thyroid remnants in patients without metastases [12,14]. In fact, normal thyroid cells widely express highly functional NIS, and a prolonged TSH stimulation is not required to obtain an effective I-131 uptake and retention in thyroid tissues, respectively [12]. Additionally, the use of rhTSH avoids THW-related hypothyroidism and the related patients’ discomfort and reduces the whole-body absorbed dose due to the faster clearance of I-131 radioactivity from the bloodstream [15,16].

Metastatic DTC cells, however, show lower density and reduced function of NIS [12,13]. As a consequence, prolonged TSH stimulation over time (i.e., a larger area under the curve of TSH stimulation) is required to increase both I-131 uptake and time of retention of I-131 in tumor cells [17,18]. Accordingly, I-131 treatment under rhTSH stimulation is generally discouraged in metastatic patients because previous studies have shown that the delivered dose to the tumor could be lower under rhTSH than under endogenous TSH stimulation, respectively. All in all, currently available guidelines generally support rhTSH as a reliable alternative to THW preparation to I-131 therapy in patients with low and intermediate-risk DTC, while THW protocols are recommended in high-risk DTC and any patient with metastatic disease [12]. Among the latters, the use of rhTSH (i.e., off-label) is restricted to patients who are either unable to elevate endogenous TSH during thyroxine withdrawal, or in whom thyroxine withdrawal is contraindicated for medical reasons [12]. Recent studies, however, reported that there should be no significant differences between rhTSH and endogenous TSH stimulation with regard to the dosimetric results and the outcome of I-131 treatment of metastatic thyroid carcinoma [19,20,21]. However, robust data are not yet available on relevant outcome measures (disease-free survival, overall survival) with rhTSH and THW in patients with metastatic DTC, and whether the two preparation methods are equivalent in the setting of metastatic radioiodine-avid DTC is still debated. Therefore, our present systematic review and meta-analysis were prompted to evaluate and compare clinical results obtained by I-131 therapy with rhTSH and THW stimulation protocols in patients with DTC metastases. In particular, the following question is to be addressed: In patients with metastatic disease, who are treated with I-131, is the patient’s preparation with rhTSH effective in terms of early response and disease progression compared with the THW protocol, respectively?

## 2. Materials and Methods

### 2.1. Protocols and Registration

Systematic review and meta-analysis were conducted according to the Preferred Reporting Items for Systematic Reviews and Meta-Analyses (PRISMA) guidelines and registered in PROSPERO (International Prospective Register of Systematic Reviews, protocol number CRD42023393178).

### 2.2. Inclusion/Exclusion Criteria

Peer-reviewed research publications were considered. The selection of eligible studies was based on the following inclusion criteria: (I) research studies, cohort studies, retrospective studies, or randomized controlled trials; (II) studies that included metastatic patients pretreated with TSH or rhTSH for 131I therapy; (III) studies that reported data about initial response to 131I therapy or progression disease.

### 2.3. Search Strategy

A systematic search was conducted om PubMed, Web of Science and Scopus from January to February 2023 with applying time or language restrictions. The literature search strategy was based on the following keywords: ((“Recombinant human TSH” OR rhTSH) AND (“thyroid hormone withdrawal” OR “off T4”)) AND (“Differentiated thyroid cancer” OR DTC OR “Papillary thyroid cancer” OR “Follicular thyroid cancer”) AND (Metastases*) AND (Radioiodine OR Iodine-131 OR I-131 OR 131I). In addition, the grey literature in the Open Grey Database (www.opengrey.eu accessed on 31 March 2023) was searched using the previously described search strategy.

### 2.4. Studies Selection

After deduplication, titles and abstract were screened independently by two authors (MLG and PPO), and ineligible articles were excluded. Subsequently, the full texts of the remaining articles were sought and assessed for eligibility by three authors (MLG, PPO, and LG). The final eligibility of each study was checked, and the reasons for exclusion were recorded. Study selection disagreement was solved by a third experienced reviewer (RG), who was consulted to reach consensus. Reasons for exclusion were written down. The process of study selection was illustrated using a PRISMA flow diagram.

Two authors (MLG and PPO) independently extracted the following information:Study characteristics: authors, publication year, country, study design, and time.Population characteristics: total sample, sample rhTSH, sample THW.Outcome evaluation.Initial response to 131I therapy after THW or rhTSH preparation.Progression disease.Onset of side-effects.

No numerical data were obtained from figures.

### 2.5. Risk of Bias (ROB)

The Newcastle-Ottawa scale was used to assess the risk of bias. Two authors independently assessed ROB. The instrument consists of eight items, four of which address selection, one of which addresses comparability, and three of which address outcome. Potential disagreements were resolved through discussion and consensus among all authors.

### 2.6. Statistical Analysis

Articles selection process and data of the included studies were reported in a narrative summary. The meta-analysis was conducted through the Mantel-Haenszel fixed effect approach. The pooled risk ratios with 95% CI were determined for both outcomes. Heterogeneity was assessed using the Cochrane Q test and the I² statistic, with a *p* value < 0.05 as an indication of statistically significant heterogeneity (Section 10.10.2 of the Cochrane Handbook for Systematic Reviews of Interventions). Heterogeneity was also examined using the L’Abbè plot: studies that deviated significantly from the effect size line were reported as outliers.

Cumulative and sensitive meta-analyses (one study removal) were also conducted to track the accumulation of evidence over time and to reduce type I errors due to small data, as well as to examine the impact of individual studies on overall prevalence results, respectively. Funnel plots and Egger’s test were also performed to investigate publication bias. Meta-analysis was performed using STATA17 (StataCorp., University Station, TX, USA).

## 3. Results

### 3.1. Search Results and Characteristics of the Studies

The literature search retrieved 450 studies: 73 articles were removed before screening due to duplication. After initial screening by title and abstract, 355 articles were excluded because they did not meet the inclusion criteria. Of the 22 remaining articles, two studies did not report the outcome of interest (i.e., first response to 131I therapy after preparation with TSH or rhTSH or disease progression), nine studies did not compare TWH and rhTSH, and one study used an overlapping sample. Ten studies met the inclusion criteria and were included in the meta-analysis (Figure 1).

### 3.2. Study Characteristics

The studies included a total of 1929 patients, of whom 953 were pre-treated with rhTSH and 976 with THW. Four studies were conducted in the United States [22,23,24,25], two in Brazil [26,27], one each in Italy [28], Portugal [20], Taiwan [29], and Canada [30]. The observation period ranged from 1993 [25] to 2019 [26], with a range of five years [29] to twenty-two years [26]. The samples consisted almost predominantly of women, except for Hugo et al.’s study, in which the male-to-female ratio was 2.2 [23]. Only three studies had a sample of more than 100 patients: 586 patients in Hugo et al. [23], 178 in Rosario et al. [27], and 647 in Tsai et al. [29]. The details of the included studies are shown in Table 1.

All included studies had a low risk of bias following a similar retrospective study design (Table 2).

Initial response was assessed using RECIST 1.1 criteria [22,25], autonomous criteria defined in previous studies [23,27,30], or ATA criteria [20]. The details of the outcome definition are shown in Table 3.

### 3.3. Initial Response to 131I Therapy after Preparation with rhTSH or THW

Six studies were included to evaluate the initial response to 131I therapy after preparation with rhTSH or THW for a total of 1042 patients [20,22,23,24,27,30]. For patients pre-treated with rhTSH, the risk ratio was 1.02 (95%CI: 0.93–1.12, *p* = 0.68, I2 = 38.42%, test of Q(5) = 8.15, *p* = 0.15) (Figure 2).

After screening for outliers by L’Abbe plot and excluding the only study that had substantial heterogeneity in effect sizes (Figure 3), the pooled risk ratio was 1.04 (95%CI: 0.94–1.14; *p* = 0.46, Q(4) = 6.03, *p* = 0.45) (Figure 4).

The cumulative meta-analysis showed an increase in the risk ratio over the years without any change in favor of a pre-treatment or the other (Figure 5).

The leave-one-out analysis remarked no effect in both groups (Figure 6). Comparisons between fixed and random effects approach revealed no changes in the pooled risk ratio (data not reported). A publication bias analysis was not performed because the number of included studies was lower than recommended for such analysis (at least 10 studies).

### 3.4. Disease Progression

All ten studies were included in the evaluation of the pooled risk ratio for disease progression for a total of 1,929 patients. The pooled risk ratio showed no effect (RR: 0.97, 95%CI: 0.92–1.03; *p* = 0.39) (Figure 7).

The Abbè Plot did not show any significant outlier (Figure 8).

The cumulative meta-analysis showed a substantial decrease in the risk ratio over time (Figure 9).

Sensitivity analysis using the leave-one-out approach revealed no relevant changes (Figure 10).

Egger’s test for small study effects revealed no significant evidence of publication bias (z = 0.75, *p* = 0.451) (Figure 11).

## 4. Discussion

In order to achieve the endogenous TSH stimulation considered adequate for radioiodine therapy (TSH > 30 mU/L), thyroidectomized patients must abstain from levothyroxine medication on average for about four weeks—which would otherwise have to be performed lifelong [31,32]. This leads to continuously increasing hypothyroid symptoms, which are perceived to varying degrees by patients—especially in the last two weeks of withdrawal. Alternatively, it is possible to bridge the first two weeks of levothyroxine withdrawal by transiently switching to liothyronine, which has a significantly shorter serum half-life. Although this avoids hypothyroidism in the first two weeks, it cannot prevent the consequences of the drop in thyroid hormone supply in the last two weeks, which then quickly sets in after discontinuation of the liothyronine [33].

The effects of transient hypothyroidism on physical and mental wellbeing, health-related quality of life, and cognitive abilities have been examined in numerous studies. A transient reduction in health-related quality of life was consistently determined [34,35,36,37,38,39]. Some studies also showed an increase in depression and anxiety, and, in other studies, this was observed mainly in patients with predisposing factors [35,36,37,39,40,41]. With previous neuropsychiatric diseases, there is a risk of exacerbation due to hypothyroidism [42]. Furthermore, a limitation of cognitive abilities is observed [34,43], which is particularly relevant for complex mental requirements. A reduction in cerebral blood flow and glucose metabolism in the short-term hypothyroid in contrast to the rhTSH-stimulated DTC patients was described as a correlate for such limitations using PET [41,44,45]. The reactivity can also be reduced, which can limit the operation of high-speed machines or the ability to drive, or at least make it appear questionable for insurance reasons—although the relevance in this regard has been assessed differently [40,46]. All of this also has economic consequences: a significantly higher time lost from work was observed in transiently hypothyroid patients [47,48]. However, complex health economic analyzes on this in comparison with rhTSH—which tend to speak for the cost-effectiveness of rhTSH stimulation—are only available to a limited extent and are also specific to the situation of the countries considered [48,49,50,51,52]. Additional advantages of using rhTSH compared to endogenous TSH stimulation could be due to lower radiation exposure from structures outside the tumorous target tissue; for example, it could be due to a reduction in the risk of salivary gland damage is reported [53,54].

For these reasons, modalities that enable the preparation of thyroid carcinoma patients for radioiodine therapy without thyroid hormone deprivation are of great relevance. Recombinant human TSH (thyrotropin-alpha, Thyrogen™, Sanofi-Aventis, Zurich, Switzerland) was approved by the FDA in 1998 (in Europe since 2000), initially for diagnostic purposes, and then it was also approved for radioiodine ablation of thyroid remnants. The drug is injected on the two days prior to radioiodine application and causes TSH levels to rise to a peak between 68 and 237 mU/L, with the peak being reached about two days after the first injection [55], without having to interrupt thyroid hormone medication. Side effects of rhTSH administration may occur, but they are less than with thyroid hormone deprivation [56]. Severe adverse events of rhTSH administration have only been described in a few case reports, e.g., a very rapid increase in size of metastases in the vicinity of critical neurological or otherwise anatomical key structures [57,58,59,60,61,62].

rhTSH is not (yet) approved for use in preparation for radioiodine therapy of a known metastatic thyroid carcinoma. Due to currently insufficient outcome data, there is no consensus as to whether the effectiveness of RIT after exogenous TSH stimulation by Thyrogen is comparable to that after endogenous TSH stimulation. Some studies found indications that the radiation doses achieved in the metastases after rhTSH stimulation are lower than with endogenous TSH stimulation [18,63,64,65]; in more recent studies, however, this potential disadvantage has been negated [19,20,21,24]. When looking at the indications for RIT in a differentiated way, it must also be emphasized that the goal of therapy, even in the case of metastases that cannot be localized, is by no means exclusively the ablation of a benign remnant of the thyroid. Rather, RIT is also seen as an adjuvant treatment to surgery to eliminate any multifocal tumor foci in the residual thyroid tissue or occult metastases—for example in lymph nodes.

So far, the guidelines of large international specialist societies only recommend the treatment of iodine-storing metastases with 131-I under exogenous rhTSH stimulation in exceptional cases (off-label use). This applies above all to patients in whom induction of hypothyroidism is associated with unacceptably high risks, or in whom adequate endogenous TSH stimulation cannot be achieved, for example due to a hypothalamo-pituitary disorder, or due to thyroid hormone-producing tumor tissue [31,32,66]. The lack of reliable data regarding the outcome of RIT in metastatic thyroid carcinoma under rhTSH stimulation compared to endogenous TSH stimulation prompted us to conduct a precise and thorough systematic review and meta-analysis to provide solid evidence for the main outcomes. Specifically, we focused on initial response and disease progression in DTC patients treated with 131I therapy after preparation with rhTSH or THW. Our results show no difference between rhTSH and THW, demonstrating a clear lack of effect size in all analyses performed. Looking more closely at the quantitative meta-analysis, the cumulative evidence shows a significant increase in the risk ratio for initial response, negating a possible significant advantage of one pretreatment over the other. The same trend was observed for disease progression: The cumulative data showed a clear indication of a lack of effect of one pretreatment over the other. This implies that concerns about the use of one or the other pretreatment should be deferred to clinical evaluations made considering patient characteristics and reduction in side effects.

This systematic review and meta-analysis have several strengths. First, no significant heterogeneity was found among the included studies, as hypothesized in the design phase of the study. This effect could be the result of a similar study design or, more likely, a similar approach, although inclusion criteria or outcome assessment were different. Second, the sensitive and cumulative analyses showed that there was no significant difference in effect size between rhTSH and THW, providing clear clinical evidence for these two pretreatment methods.

Some limitations should be disclosed as well. First, all included studies have a retrospective study design, which may have increased the risk of selection bias and resulting bias in outcome assessment. Second, only three of the 10 selected studies reported the frequency of side effects, a circumstance that prevented us from conducting an accurate meta-analysis on side effects. Third, although the aim of all included studies was to investigate the effect of rhTSH or THW in patients with metastases, the inclusion criteria seemed broad at baseline. Namely, while some studies specified the presence of metastases as inclusion criteria at the diagnosis, other studies included patients who did not have metastases at diagnosis but developed them during the follow-up phase. Regarding inclusion criteria, the included studies considered more or less heterogeneous samples in terms of DTC diagnosis: some studies focused only on PTC, the most common form of DTC, whereas other studies also included different forms of DTC (i.e., FTC). In addition, different definitions of disease progression were used. The definition of disease progression and that of structural or biochemical incomplete response were often used with the same meaning. Finally, no prospective randomized studies comparing the two preparation modalities are currently available, making them warranted in the near future.

## 5. Conclusions

The cumulative data of our systematic review and meta-analysis indicated of a lack of significant effect of rhTSH or THW pretreatment over the other on the effectiveness of I-131 therapy of metastatic DTC. This implies that concerns about the use of one or the other pretreatment should be deferred to clinical evaluations made considering patient characteristics and reduction in side effects.

## Figures and Tables

**Figure 1 cancers-15-02510-f001:**
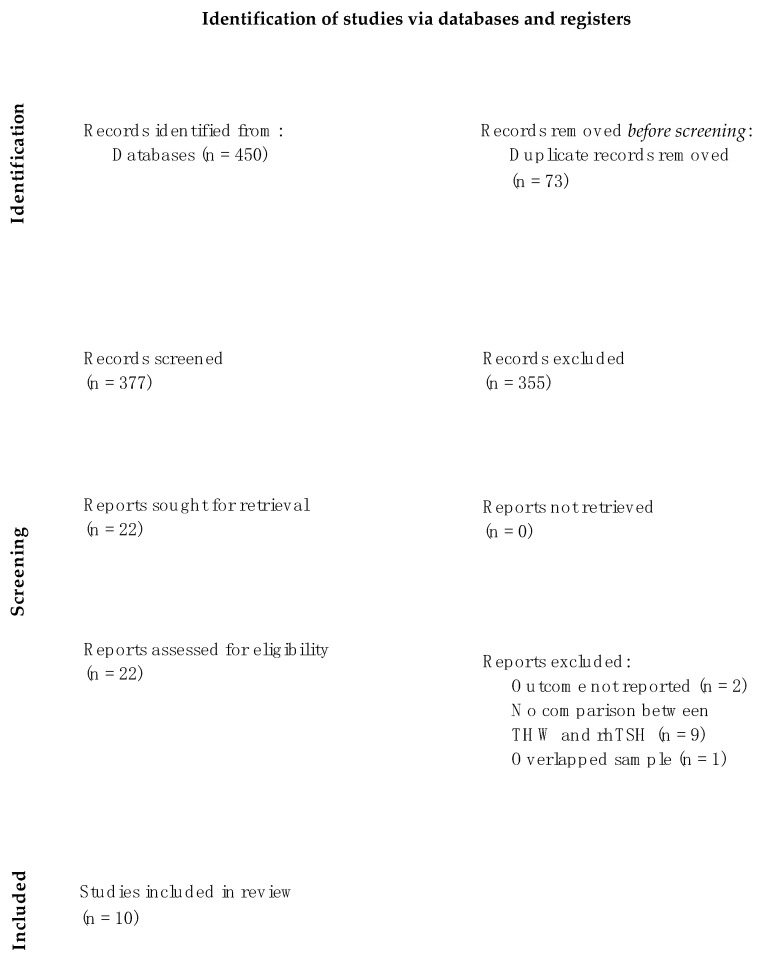
PRISMA flowchart.

**Figure 2 cancers-15-02510-f002:**
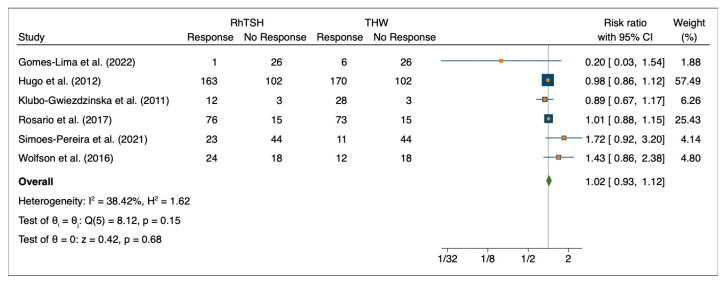
Forest Plot—initial response to 131I therapy after pre-treatment with rhTSH or THW (n = 6).—Gomes-Lima et al. (2022) [22], Hugo et al. (2012) [23], Klubo-Gwiezdzinska et al. (2011) [24], Rosario et al. (2017) [27], Simoes-Pereira et al. (2021) [20], Wolfson et al. (2016) [30].

**Figure 3 cancers-15-02510-f003:**
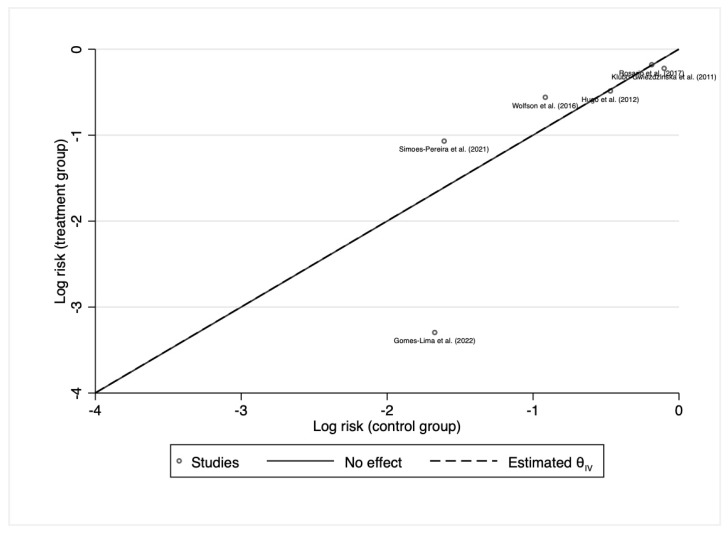
L’Abbe Plot for initial response to 131I therapy after pre-treatment with rhTSH or THW. Gomes-Lima et al. (2022) [22], Hugo et al. (2012) [23], Klubo-Gwiezdzinska et al. (2011) [24], Rosario et al. (2017) [27], Simoes-Pereira et al. (2021) [20], Wolfson et al. (2016) [30].

**Figure 4 cancers-15-02510-f004:**
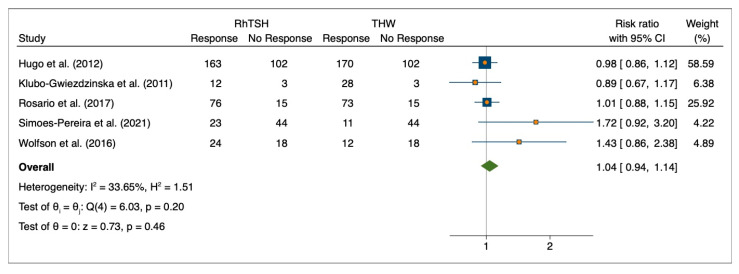
Forest plot for initial response to 131I therapy after pre-treatment with rhTSH or THW excluding outliers (n = 5). Hugo et al. (2012) [23], Klubo-Gwiezdzinska et al. (2011) [24], Rosario et al. (2017) [27], Simoes-Pereira et al. (2021) [20], Wolfson et al. (2016) [30].

**Figure 5 cancers-15-02510-f005:**
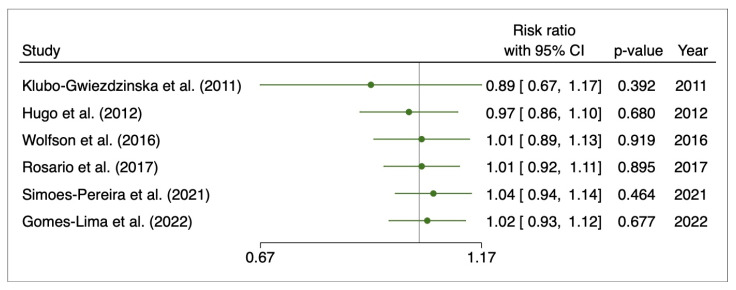
Cumulative meta-analysis for initial response to 131I therapy after pre-treatment with rhTSH or THW. Gomes-Lima et al. (2022) [22], Hugo et al. (2012) [23], Klubo-Gwiezdzinska et al. (2011) [24], Rosario et al. (2017) [27], Simoes-Pereira et al. (2021) [20], Wolfson et al. (2016) [30].

**Figure 6 cancers-15-02510-f006:**
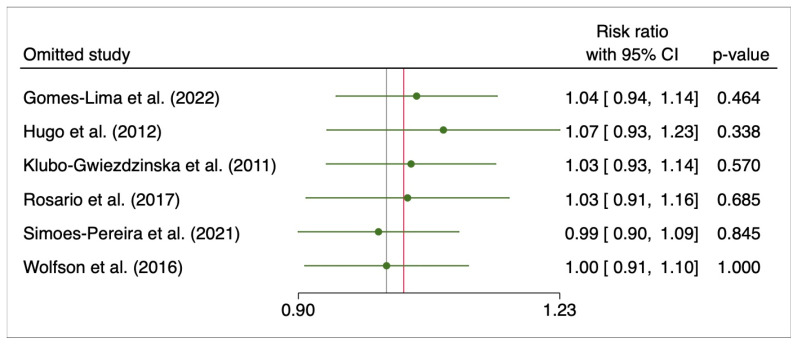
Leave-one-out for initial response to 131I therapy after pre-treatment with rhTSH or THW. Gomes-Lima et al. (2022) [22], Hugo et al. (2012) [23], Klubo-Gwiezdzinska et al. (2011) [24], Rosario et al. (2017) [27], Simoes-Pereira et al. (2021) [20], Wolfson et al. (2016) [30].

**Figure 7 cancers-15-02510-f007:**
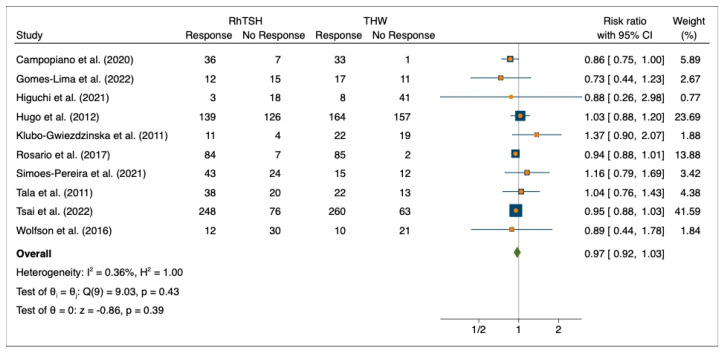
Forest plot for progression disease. Campopiano et al. (2020) [28], Gomes-Lima et al. (2022) [22], Higuchi et al. (2021) [26], Hugo et al. (2012) [23], Klubo-Gwiezdzinska et al. (2011) [24], Rosario et al. (2017) [27], Simoes-Pereira et al. (2021) [20], Tala et al. (2011) [25], Tsai et al. (2023) [29], Wolfson et al. (2016) [30].

**Figure 8 cancers-15-02510-f008:**
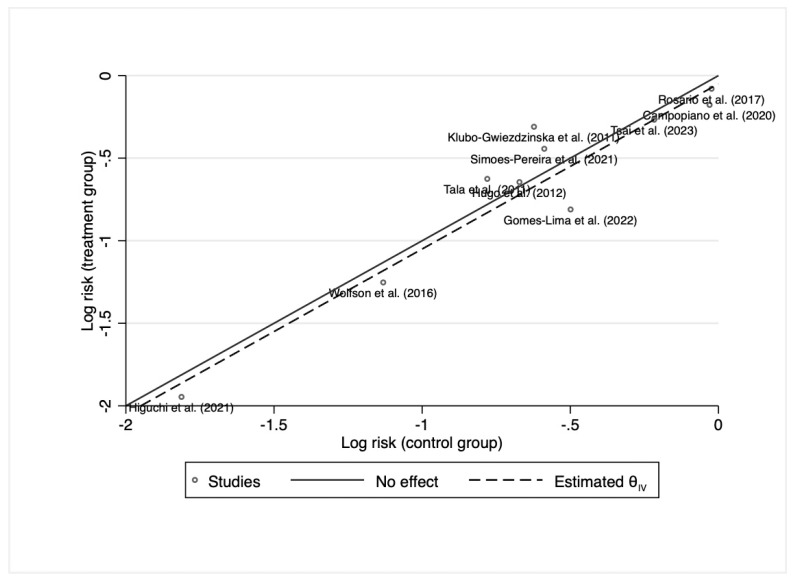
L’Abbè Plot for progression disease. Campopiano et al. (2020) [28], Gomes-Lima et al. (2022) [22], Higuchi et al. (2021) [26], Hugo et al. (2012) [23], Klubo-Gwiezdzinska et al. (2011) [24], Rosario et al. (2017) [27], Simoes-Pereira et al. (2021) [20], Tala et al. (2011) [25], Tsai et al. (2023) [29], Wolfson et al. (2016) [30].

**Figure 9 cancers-15-02510-f009:**
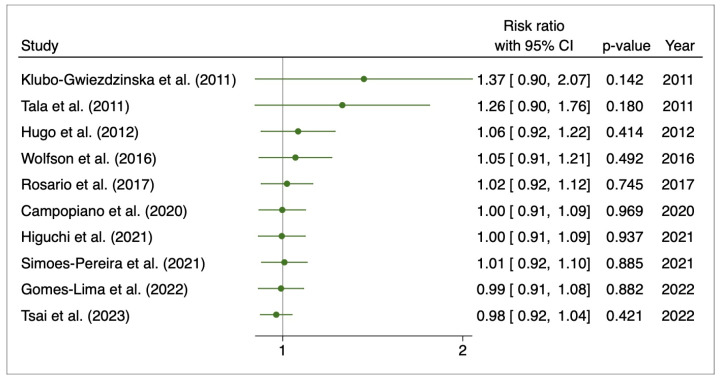
Cumulative meta-analysis for progression disease. Campopiano et al. (2020) [28], Gomes-Lima et al. (2022) [22], Higuchi et al. (2021) [26], Hugo et al. (2012) [23], Klubo-Gwiezdzinska et al. (2011) [24], Rosario et al. (2017) [27], Simoes-Pereira et al. (2021) [20], Tala et al. (2011) [25], Tsai et al. (2023) [29], Wolfson et al. (2016) [30].

**Figure 10 cancers-15-02510-f010:**
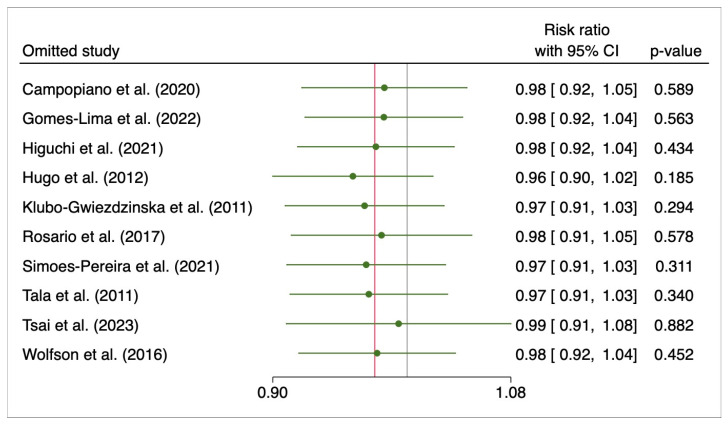
Leave-one-out for progression disease. Campopiano et al. (2020) [28], Gomes-Lima et al. (2022) [22], Higuchi et al. (2021) [26], Hugo et al. (2012) [23], Klubo-Gwiezdzinska et al. (2011) [24], Rosario et al. (2017) [27], Simoes-Pereira et al. (2021) [20], Tala et al. (2011) [25], Tsai et al. (2023) [29], Wolfson et al. (2016) [30].

**Figure 11 cancers-15-02510-f011:**
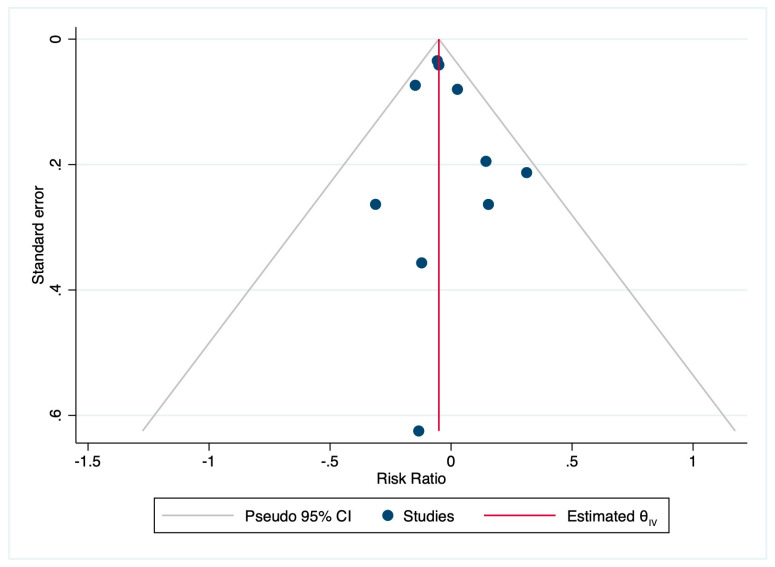
Funnel Plot—publication bias.

**Table 1 cancers-15-02510-t001:** Studies Characteristics.

Author	Design	Country	Period	Sample			Age (years)	Sex (Male/Female)	DTC		Follow-Up	
				Total	rhTSH	THW			rhTSH	TWH	rhTSH	TWH
Campopiano et al. (2020) [28]	RS	Italy	2001–2017	77	43	34	rhTSH: 48 ± 20	rhTSH: 21/43	PTC: 35; FTC: 8	PTC: 29; DTC: 5	45 ± 46 mo	48 ± 27 mo
							THW: 41 ± 17	THW:16/34				
Gomes-Lima et al. (2022) [22]	RS	USA	1996–2017	55	27	28	rhTSH: 59 (47.5–65.5)	rhTSH: 5/22	PTC: 19; FTC: 6; HCC: 1; PD: 1	PTC: 21; FTC: 4;HCC: 3; PD: 0	4.2 years (3.3–5.5)	6.9 yr (4.2–11.6)
							THW: 41 (30.9–63.5)	THW: 11/17				
Higuchi et al. (2021) [26]	RS	Brazil	1997–2019	70	21	49	rhTSH: 63 (24–83)	rhTSH: 9/12	PTC: 13; FTC: 5; HCC: 1; PD: 2	PTC: 31; FTC: 9; HCC: 5; PD: 4	61 mo (19–149)	88 mo (13–241)
							THW: 63 (31–79)	THW: 20/29				
Hugo et al. (2012) [23]	RS	USA	1994–2004	586	265	321	rhTSH: 46 ± 15	rhTSH: 180/85	PTC: 225; PD: 12; FTC: 10; HCC: 18	PTC: 271; PD: 26; FTC: 13; HCC: 11	8.1 ± 3.3 years	8.8 ± 3.6 yr
							THW: 47 ± 15	THW: 221/100				
Klubo-Gwiezdzinska et al. (2011) [24]	RS	USA	1996–2009	56	15	41	rhTSH: 62.4 ± 12.6	rhTSH: 4/11	PTC: 36; FTC: 8; HCC: 7; Other: 5		72 ± 36.2 mo	
							THW: 48.8 ± 18.2	THW: 21/20				
Rosario et al. (2017) [27]	RS	Brazil	2006–2014	178	91	87	rhTSH: 47 (18–76)	rhTSH:23/68	Only PTC		64 mo (24–118)	68 mo (18–118)
							THW: 48 (18–72)	THW: 21/66				
Simoes-Pereira et al. (2021) [20]	RS	Portugal	2006–2018	94	67	27	rhTSH *: 65.5 (22–85)	rhTSH: 27/41	Only PTC		68 mo * (8–332)	120 mo * (9–332)
							THW *: 58.9 (20–77)	THW: 7/20				
Tala et al. (2011) [25]	RS	USA	1993–2010	93	58	35	rhTSH: 60 (20–89)	rhTSH: 28/30	PTC: 29; FTC. 5; HCC: 4; PD: 16; Other: 4	PTC: 20; FTC. 6; PD: 8; Other: 1	3.4 mo (1.3–10.3)	6.9 mo (1.4–17.1)
							THW: 56 (24–80)	THW: 15/20				
Tsai et al. (2023) [29]	RS	Taiwan	2013–2018	647	324	323	rhTSH: 49.66 ± 14.40	rhTSH: 72/252	PTC: 307; HCC: 2; FTC: 15	PTC: 306; HCC: 5; FTC: 12	NR	NR
							THW: 49.29 ± 13.06	THW: 88/235				
Wolfson et al. (2016) [30]	RS	Canada	2007–2018	73	42	31	rhTSH: 45.7 ± 16.2	rhTSH: 17/25	PTC: 41; PD: 1	PTC: 29; PD: 1; Other: 1	6.8 ± 2.1	8.6 ± 2.4
							THW: 38.2 ± 12.4	THW: 6/25				

Legend *: IQR = Interquartile Range, PTC: Papillary Thyroid Cancer, FTC: Follicular Thyroid Cancer, HCC: Hürthle cell carcinoma, PD: Progression Disease—RS: Retrospective Study.

**Table 2 cancers-15-02510-t002:** Risk of Bias—Newcastle Ottawa Scale (NOS).

Authors	Selection	Comparability	Outcome
Campopiano et al. (2020) [28]	****	*	**
Gomes-Lima et al. (2022) [22]	****	*	**
Higuchi et al. (2021) [26]	****	*	**
Hugo et al. (2012) [23]	****	*	**
Klubo-Gwiezdzinska et al. (2011) [24]	****	*	**
Rosario et al. (2017) [27]	****	*	**
Simoes-Pereira et al. (2021) [20]	****	*	**
Tala et al. (2011) [25]	****	*	**
Tsai et al. (2023) [29]	****	*	**
Wolfson et al. (2016) [30]	****	*	**

NOS is based on a star rating system. Each star indicates whether or not the specific item of the scale was identified in the respective study. The scale consists of eight items (four for selection, maximum four stars, one for comparability, maximum two stars).

**Table 3 cancers-15-02510-t003:** Outcome to assess response and/or progression disease.

Author	Outcome
Campopiano et al. (2020) [28]	RECIST 1.1 criteria
Gomes-Lima et al. (2022) [22]	RECIST 1.1 criteria
Higuchi et al. (2021) [26]	Stable disease = no structural progression in the last year of follow-up; Disease progression = an increase or appearance of a new structural lesion in the last year of follow-up
Hugo et al. (2012) [23]	Best response to initial therapy (first two years of follow-up): excellent response (suppressed and stimulated Tg < 1 ng/mL; neck US with no evidence of disease and no other cross sectional or functional evidence of disease); acceptable response (suppressed Tg < 1 ng/mL with stimulated Tg 1–10 ng/mL or non-specific findings on neck US or other imaging); incomplete response (suppressed Tg > 1 ng/mL, stimulated Tg > 10 ng/mL, or structural evidence of persistent disease). Clinical status at the time of last follow-up: no evidence of disease (suppressed Tg < 1 ng/mL or no detectable anti-Tg antibody and no structural or functional evidence of disease); persistent disease (suppressed Tg values > 1 ng/mL; stimulated Tg values > 2 ng/mL; evidence of persistent disease in structural or function imaging or biopsy-proven disease); recurrent disease (suppressed Tg > 1 ng/mL or structural or functional evidence of disease identified following a period of no evidence of disease)
Klubo-Gwiezdzinska et al. (2011) [24]	RECIST 1.1 criteria
Rosario et al. (2017) [27]	(i) Rate of excellent response to therapy, i.e., nonstimulated Tg ≤ 0.2 ng/mL, with negative TgAb and negative neck US, one year after RAI (1–4); (ii) structural disease one year after RAI; (iii) structural or biochemical (nonstimulated Tg > 1 ng/mL, with increment) recurrence during follow-up, and (iv) percentage of patients without disease in the last assessment, i.e., nonstimulated Tg < 1 ng/mL and no evidence of structural disease
Simoes-Pereira et al. (2021) [20]	(i) Malignant tissue that does not concentrate RAI on a post-RAIT WBS; (ii) tumor tissue that loses the ability to concentrate RAI after previous evidence of RAI-avid disease; (iii) RAI uptake that is concentrated in some lesions, but not in others; (iv) metastatic disease that progresses despite significant concentration of RAI; and (v) 600 mCi of cumulative RAIT
Tala et al. (2011) [25]	Progression disease
Tsai et al. (2023) [29]	Excellent treatment response (non-stimulated Tg levels of <0.2 ng/mL, undetectable TgAb, and negative imaging on neck ultrasonography and DxWBS); biological incomplete response (abnormal Tg levels, rising anti-Tg antibody levels, and lack of localizable disease on imaging); structural incomplete response (persistent, newly loco-regional, or distant metastases revealed on thyroid ultrasonography or other imaging); indeterminate response (non-specific biochemical or structural findings, which could not be classified as either benign or malignant)
Wolfson et al. (2016) [30]	Response to initial treatment: excellent (both suppressed and stimulated Tg were <1 ng/mL and no evidence of disease on neck ultrasound, whole body iodine scan, or CT scan); acceptable response (suppressed Tg < 1 ng/mL, stimulated Tg 1–10 ng/mL, and/or equivocal findings on diagnostic imaging); incomplete response (suppressed Tg > 1 ng/mL, stimulated Tg > 10 ng/mL, and/or evidence of persistent disease on diagnostic imaging). Final outcome: no evidence of disease (suppressed Tg < 1 ng/mL, no detectable anti-Tg antibody, and no structural evidence of disease on clinical examination or radiological studies); persistent disease (suppressed Tg values > 1 ng/mL, stimulated Tg values > 2 ng/ mL, and/or evidence of persistent disease in structural or functional imaging); recurrent disease (suppressed Tg > 1 ng/mL and/or structural or functional evidence of disease identified following a period of no evidence of disease)

## Data Availability

No new data were created. All primary information included in the meta-analysis is reported in the related forest plots.

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
