# Peer review of "Thyroid Hormone Withdrawal versus Recombinant Human TSH as Preparation for I-131 Therapy in Patients with Metastatic Thyroid Cancer: A Systematic Review and Meta-Analysis"

_cancers, 2023, doi:10.3390/cancers15092510_

Round 1

Reviewer 1 Report

The authors address an important and  debated clinical issue regarding the use of human recombinant TSH in the treatment of metastases of differentiated thyroid cancer.

Whether radiometabolic therapy performed in conditions of hypothyroidism is cost-effective, (meaning as costs not only economic) or can be safely replaced by pretreatment with rTSH not only for  131-I ablation but also for metastases is certainly a debated topic and for which there is still no consensus.

The authors have carried out a review and meta-analysis of the available literature and conclude that there are no substantial differences in terms of efficacy between the two modes of radioiodine administration, supporting the arguments in favor of a diversified use for the different clinical conditions of the patient or other conditions but substantially refuting a different efficacy.

The work is interesting, very well structured and fluenty to read, and therefore in my opinion certainly worthy of publication, despite, as also reported by the authors, its strength is decidedly limited by the retrospective nature of the available studies.

It would be useful for the authors in the discussion to point out the need for the absence of randomized trials comparing the two modalities.

Author Response

Reviewer 1

The authors address an important and  debated clinical issue regarding the use of human recombinant TSH in the treatment of metastases of differentiated thyroid cancer.

Whether radiometabolic therapy performed in conditions of hypothyroidism is cost-effective, (meaning as costs not only economic) or can be safely replaced by pretreatment with rTSH not only for 131-I ablation but also for metastases is certainly a debated topic and for which there is still no consensus.

The authors have carried out a review and meta-analysis of the available literature and conclude that there are no substantial differences in terms of efficacy between the two modes of radioiodine administration, supporting the arguments in favor of a diversified use for the different clinical conditions of the patient or other conditions but substantially refuting a different efficacy.

The work is interesting, very well structured and fluenty to read, and therefore in my opinion certainly worthy of publication, despite, as also reported by the authors, its strength is decidedly limited by the retrospective nature of the available studies.

We thanks the Rev1 for her/his positive words and appreciation of our paper.

It would be useful for the authors in the discussion to point out the need for the absence of randomized trials comparing the two modalities.

Thanks for this remark. We added a sentence in the Discussion to remind this relevant point.

Reviewer 2 Report

General comment

This paper by Giovanella et al is a systematic review and meta-analysis on the comparison between thyroid hormone withdrawal (THW) and recombinant human TSH (Thyrogen) as preparation for 131-I therapy in patients with metastatic thyroid cancer. This paper is clearly written, the search of available literature is accurate and the meta-analytic methodology complete. My main (and only) concern is that there are presently very few studies (all retrospective and not easily comparable) to allow a full meta-analytical evaluation to draw solid conclusions which could be translated to clinical guidelines. This concept, on the other hand, is clearly acknowledged by the authors. There a few (mostly typographical/page setting) imprecisions which may hamper the reading of the paper, as detailed below (specific comments)

Specific comments

Abstract line 33: “131I therapy”; 131-iodine is abbreviated throughout all paper as I-131: please, use always the same abbreviation

Figure 1: Please increase the dimension of the text fonts and add lines for boxes and connectors

Table 1. Avoid to break the table as presently shown (Headers and first row in one page and the remaining table in the subsequent page): try to show the entire table in the same page. The text of the headers is presently difficult to read, try to improve

Table 2 (Risk of Bias): A reference to the Newcastle Ottawa Scale should be provided (in the Table footer or in the paragraph ”Risk of bias” in the “Materials and Methods” section). The meaning of *, ** and **** symbols is unclear and should be specified in the Table 2 footer.

Table 3: again, as for Table 2, the break within the last row of the Table, i.e. that related to the paper by Wolfson et al (2016) hampers the reading of the table. Try to put all the table in one page.

Figure 2 is not easily readable (low definition and fonts too small)

Figure 3 is not easily readable (low definition and fonts really too small)

Figures 4 and 5: try to improve definition (font and symbols dimension OK)

Figure 7: is not easily readable (low definition and fonts too small)

Figure 8: is not easily readable (low definition and fonts too small)

Figure 9: The decrease in the risk ratio over time is defined “significant” in the text, but the level of statistical significance is not provided in the text or shown in Fig. 9

Author Response

Reviewer #2

General comment

This paper by Giovanella et al is a systematic review and meta-analysis on the comparison between thyroid hormone withdrawal (THW) and recombinant human TSH (Thyrogen) as preparation for 131-I therapy in patients with metastatic thyroid cancer. This paper is clearly written, the search of available literature is accurate and the meta-analytic methodology complete. My main (and only) concern is that there are presently very few studies (all retrospective and not easily comparable) to allow a full meta-analytical evaluation to draw solid conclusions which could be translated to clinical guidelines. This concept, on the other hand, is clearly acknowledged by the authors. There a few (mostly typographical/page setting) imprecisions which may hamper the reading of the paper, as detailed below (specific comments)

Specific comments

Abstract line 33: “131I therapy”; 131-iodine is abbreviated throughout all paper as I-131: please, use always the same abbreviation

Figure 1: Please increase the dimension of the text fonts and add lines for boxes and connectors

Table 1. Avoid to break the table as presently shown (Headers and first row in one page and the remaining table in the subsequent page): try to show the entire table in the same page. The text of the headers is presently difficult to read, try to improve 

Authors’ response: We have reported the table in the same page.

Table 2 (Risk of Bias): A reference to the Newcastle Ottawa Scale should be provided (in the Table footer or in the paragraph ”Risk of bias” in the “Materials and Methods” section). The meaning of *, ** and **** symbols is unclear and should be specified in the Table 2 footer.

Authors’ response: We have added the explanation of the NOS star rating system.

Table 3: again, as for Table 2, the break within the last row of the Table, i.e. that related to the paper by Wolfson et al (2016) hampers the reading of the table. Try to put all the table in one page.

Authors’ response: We have inserted the table in one page.

Figure 2 is not easily readable (low definition and fonts too small)

Figure 3 is not easily readable (low definition and fonts really too small)

Figures 4 and 5: try to improve definition (font and symbols dimension OK)

Figure 7: is not easily readable (low definition and fonts too small)

Figure 8: is not easily readable (low definition and fonts too small)

Authors’ response: The reported problem is not present in the manuscript. It may be a problem related to the final layout generated by the automatic pagination. For your convenience, we are providing a PDF copy of our manuscript in which the font and low-resolution problems do not appear to be present.

Figure 9: The decrease in the risk ratio over time is defined “significant” in the text, but the level of statistical significance is not provided in the text or shown in Fig. 9

Authors’ response: We have explained better the figure reading changing significant with substantial. It is not a statistical significance, but it a real trend in the that it could be extracted simply reading the cumulative meta-analysis figure.

Reviewer 3 Report

Review Manuscript IDcancers-2351420

Type of manuscript: Review

Title: Thyroid Hormone Withdrawal versus Recombinant Human TSH as Preparation for I-131 Therapy in Patients with Metastatic Thyroid Cancer. A Systematic Review and Meta-analysis

Authors: Luca Giovanella * , Maria Luisa Garo , Alfredo Campenni’ , Petra Petranovic , Rainer Goerges

It is a very interesting manuscript, with solid documentation and excellent statistical analysis.

The results of this review should have a considerable impact.

Strengths of the manuscript:

1.      The subject-it has been more than 7 years since the ATA guideline, ESMO guideline regarding the diagnostic and treatment of differentiated thyroid cancer avoided clarifying the subject 

2.     The solid documentation of the published data and references

3.     The statistical analysis

Minor consideration and recommendation

1.     Please consider to rephrase the following sentences regarding the “side effects”, both in the simple summary and abstract lines 24 and 44/page 1,  and line 331 /page 13, line 360/page 14.

“This implies that concerns about the use of one or the other pretreatment should be deferred to clinical evaluations made considering patient characteristics and reduction in side effects”.

2.     The authors mentioned as one of the limitation of the study, the fact that 

“The metanalysis did not assessed the frequency of side effects… only three of the 10 studies reported such information, a circumstance that prevented us from conducting an accurate meta-analysis on side effects”.

Author Response

Reviewer 3

It is a very interesting manuscript, with solid documentation and excellent statistical analysis. The results of this review should have a considerable impact.

Strengths of the manuscript:

1.he subject-it has been more than 7 years since the ATA guideline, ESMO guideline regarding the diagnostic and treatment of differentiated thyroid cancer avoided clarifying the subject 

2.The solid documentation of the published data and references

3.The statistical analysis

We thanks the reviewer for her/his positive words and appreciation of our manuscript.

Minor consideration and recommendation

  1. Please consider to rephrase the following sentences regarding the “side effects”, both in the simple summary and abstract lines 24 and 44/page 1,  and line 331 /page 13, line 360/page 14.

“This implies that concerns about the use of one or the other pretreatment should be deferred to clinical evaluations made considering patient characteristics and reduction in side effects”.

 Thanks for this suggestion. We rephrased accordingly.

  1. The authors mentioned as one of the limitation of the study, the fact that 

“The metanalysis did not assessed the frequency of side effects… only three of the 10 studies reported such information, a circumstance that prevented us from conducting an accurate meta-analysis on side effects”.

Thanks for this remark, we rephrased as suggested.